# Marshall Stability Prediction with Glass and Carbon Fiber Modified Asphalt Mix Using Machine Learning Techniques

**DOI:** 10.3390/ma15248944

**Published:** 2022-12-14

**Authors:** Ankita Upadhya, Mohindra Singh Thakur, Mohammed Saleh Al Ansari, Mohammad Abdul Malik, Ahmad Aziz Alahmadi, Mamdooh Alwetaishi, Ali Nasser Alzaed

**Affiliations:** 1Department of Civil Engineering, Shoolini University, Solan 173229, Himachal Pradesh, India; 2Department of Chemical Engineering, College of Engineering, University of Bahrain, Zallaq P.O. Box 32038, Bahrain; 3Engineering Management Department, College of Engineering, Prince Sultan University, P.O. Box 66833, Riyadh 11586, Saudi Arabia; 4Department of Electrical Engineering, College of Engineering, Taif University, P.O. Box 11099, Taif 21944, Saudi Arabia; 5Department of Civil Engineering, College of Engineering, Taif University, P.O. Box 11099, Taif 21944, Saudi Arabia; 6Department of Architecture Engineering, College of Engineering, Taif University, P.O. Box 11099, Taif 21944, Saudi Arabia

**Keywords:** asphalt modified mix, artificial neural network, support vector machines, gaussian processes, M5P tree, multiple linear regression, Marshall stability, glass fiber, carbon fiber, hybrid mix

## Abstract

Pavement design is a long-term structural analysis that is required to distribute traffic loads throughout all road levels. To construct roads for rising traffic volumes while preserving natural resources and materials, a better knowledge of road paving materials is required. The current study focused on the prediction of Marshall stability of asphalt mixes constituted of glass, carbon, and glass-carbon combination fibers to exploit the best potential of the hybrid asphalt mix by applying five machine learning models, i.e., artificial neural networks, Gaussian processes, M5P, random tree, and multiple linear regression model and further determined the optimum model suitable for prediction of the Marshall stability in hybrid asphalt mixes. It was equally important to determine the suitability of each mix for flexible pavements. Five types of asphalt mixes, i.e., glass fiber asphalt mix, carbon fiber asphalt mix, and three modified asphalt mixes of glass-carbon fiber combination in the proportions of 75:25, 50:50, and 25:75 were utilized in the investigation. To measure the efficiency of the applied models, five statistical indices, i.e., coefficient of correlation, mean absolute error, root mean square error, relative absolute error, and root relative squared error were used in machine learning models. The results indicated that the artificial neural network outperformed other models in predicting the Marshall stability of modified asphalt mix with a higher value of the coefficient of correlation (0.8392), R^2^ (0.7042), a lower mean absolute error value (1.4996), and root mean square error value (1.8315) in the testing stage with small error band and provided the best optimal fit. Results of the feature importance analysis showed that the first five input variables, i.e., carbon fiber diameter, bitumen content, hybrid asphalt mix of glass-carbon fiber at 75:25 percent, carbon fiber content, and hybrid asphalt mix of glass-carbon fiber at 50:50 percent, are highly sensitive parameters which influence the Marshall strength of the modified asphalt mixes to a greater extent.

## 1. Introduction

Scientists and researchers are continuously working on finding ways and means to enhance the performance of flexible pavement for highways for which they have used glass and carbon fibers in asphalt mix to improve its structural performance. Carbon fibers are strong enough due to their high tensile and modulus of elasticity properties, whereas glass fibers are cheap and useful in imparting crack resistance. However, there is not enough literature available on the performance of the hybrid asphalt mix blended with glass and carbon fibers together. Therefore, it was necessary to investigate the performance of such hybrid asphalt mixes with glass and carbon fibers to obtain strength properties as good as that of carbon fiber-modified asphalt mix. The current study focused on the prediction of the Marshall stability of asphalt mixes comprising glass, carbon, and glass-carbon combination fibers by applying five machine learning models, i.e., artificial neural networks, Gaussian processes, M5P, random tree, and multiple linear regression models; furthermore, we determined the optimum model suitable for prediction of the Marshall stability in hybrid asphalt mixes. Equally important was to determine the suitability of each mix for flexible pavements. The criteria for selecting the aforementioned five machine learning techniques was their veracity in their functions. Therefore, the impact of said fibers on the Marshall stability of the asphalt mix to be used in flexible pavements is of paramount significance. Pavement cracking can be caused by the insufficient flexural and tensile strength of asphalt concrete. Cracking is a typical kind of pavement damage that shortens the life of roads [1]. Asphalt pavements often undergo fatigue cracking, which decreases the pavement service life and raises the costs of both maintenance and driving. It is of the utmost importance to discover strategies for reducing the rate of asphalt pavement deterioration and increasing its useful life duration. Many studies have been conducted to improve road surface properties that can provide a more pleasant ride and increase durability [2].

Many techniques and the use of a variety of additives have been proposed to increase the anti-cracking capabilities of asphalt pavements. Due to tremendous improvement benefits, low cost, and ease of construction, fibers have risen in popularity as a modifier of asphalt mixtures [3]. Fibers increase the ductility of asphalt mixes by enhancing their elasticity, resistivity, toughness, and distortion ability [4]. There are a variety of fibers that can enhance the quality of road pavements, including glass fiber, polyester fiber, sisal fiber, basalt fiber, polyester, carbon fiber, etc. [5]. Studies have shown that adding glass fiber, polypropylene fiber lime, and nano-silica powder to asphalt pavements improves the rutting capabilities and leads to increased direct and indirect tensile strength [6]. Adding polypropylene (PP) and glass fiber (GF) to the asphalt mix improves the mechanical and durability features of asphalt concrete pavements while reducing the draining effect of the asphalt material [7]. The addition of fibers to asphalt increases the mixture’s quality and durability, resulting in reduced road maintenance costs and a longer lasting road. Fibers added to asphalt mixes improve mechanical performance by creating a three-dimensional network that gives hot-mix asphalt more strength and a stronger grip on roads [8]. In the study [9], it was discovered that adding glass fibers to a stone mastic asphalt mix improves the mix properties, and that using 0.4% fibers with 6.0% binder concentration results in greater stability and low drain-down values. Raveling resistance and moisture vulnerability are much improved in epoxy asphalt mixes using glass fiber [10]. Marshall and bitumen mixture performance are significantly impacted by the length of glass fiber used [11]. Glass-fiber-reinforced asphalt concrete (GFRAC) outperformed unreinforced asphalt in the Marshall stability test [12]. The addition of glass fiber and diatomite to the conventional asphalt mixture improves its fatigue qualities and rutting resistance [13]. Adding (0.30%) lignin and glass fiber (GF) increased the water stability and the quality of the asphalt mix [14].

Researchers specializing in the area of machine learning (ML) hypothesized that a combination of different ML techniques would be essential to overcome the problem of diversity and complexity in learning scenarios [15,16]. Many analyses have been conducted to solve engineering problems and mathematical ideas that will help engineers create engineered tools such as structures, machines, items, and processes [17]. Researchers have paid a lot of attention to the use of artificial intelligence (AI) for determining the mechanical behavior of asphalt concrete materials because of its easiness and reliability. Assessing the impact of non-linear data and non-factors on recurrent neural networks is a common application of machine intelligence techniques such as gaussian process regression, random forest, random tree, M5P tree, gene expression program (GEP), support vector machine (SVM), Gaussian process (GP), fuzzy logic, and ANFIS [16,18,19,20,21,22,23,24,25,26]. The ANN approach [18] was implemented to predict the sustainability of asphalt concrete at various temperatures, which is better at predicting non-linear data. The study [18] found the efficacy of the generated models and further compared them with the most widely used dynamic modulus prediction and ANN models. The M5P-based models outperform other applied models [19]. In addition, the logarithmic change in the values of elastic stiffness considerably enhances the model performance. The ANN analytic techniques are quick and correct in predicting bending and critical conditions of pavement structures exposed to standard traffic stresses [20]. In a study [21], ML techniques were applied to predict Marshall characteristics, i.e., MS, permanent deformation, and several air voids of asphalt pavement and surface course. On the other hand, study [22] examined the numerical and experimental data of glass-fiber-reinforced polymer (GFRP) mixes. The application of the Gaussian process regression (GPR) technique showed more accuracy in estimating the rutting characteristic [23]. Additionally, the fatigue parameter can be predicted more precisely using unaged input variables. In the study performed by [24], an ANN model was generated to predict the fracture toughness and rutting pavement thickness of reinforced asphalt and showed more effectiveness of the model. The permeability coefficient was estimated using M5P and GP which demonstrated more accuracy in prediction [25]. The study performed by [26] implemented SVMs to improve the asphalt-pavement resilience modulus and structural performance indicators of pavement materials. The authors of [27] developed gene expression analysis and several dimensionality reduction techniques based on matrix factorization. The results show that it is effective and productive for the gene selection function. The study [28] implemented robust graph regularization non-negative matrix factorization for attributed networks incorporating two sources of data, namely network topology and node properties; the results show that the performance of the prediction is greatly improved when attributed and topological information is combined. It was found in the study [29] that by using a search-based technique and a late fusion strategy, appropriate tags are proposed for each test data throughout the prediction phase. The prior studies related to machine learning techniques are shown in Table 1.

The objective of this study was to predict Marshall stability (MS) with ten input parameters, i.e., BC, GF, 75GF:25CF, 50GF:50CF, 25GF:75CF, CF, VG, FL, FD Glass Fiber, and FD Carbon Fiber by applying five machine learning models, i.e., artificial neural networks, Gaussian processes, M5P, random tree, and multiple linear regression models; a further objective was to determine the optimum model suitable for prediction of the Marshall stability for the same set of input variables. It was equally important to determine the suitability of each mix for flexible pavements by performing the sensitivity analysis.

## 2. Machine Learning Models

### 2.1. ANN Model

ANNs are based on the structure and function of biological brains (representing the number of hidden neurons and one output neuron). The weighted connection between two layers stands for the number of nodes in each layer [39]. For a more practical ANN network, we can utilize iterative learning. By using a black box method, the prediction equation is obscured. Each layer’s contribution to the network’s data flow is noted. In the context of training, epochs are cycles of data collection. Training time for ANNs grows exponentially with the size of the dataset [40]. Sigmoid, biased, and linear output layers can approximate finitely discontinuous functions. Sigmoid functions output the products and weights of the preceding neurons’ outputs. Division and exponent math make the sigmoid function difficult to directly implement in circuits. Sigmoid is included in neural networks and deep-learning systems in several ways [16]. Figure 1 depicts the ANN structure.

### 2.2. GP Model

GP, a stochastic process, follows a multivariate normal distribution for finite random variables. GP interprets kernel models and kernel machines. Gaussian process log-marginal-likelihood maximizes kernel hyperparameters in regressor fitting (LML). All finite random variables are jointly normal. For GP Bayesian non-parametric modelling, correlation drives this “non-parametric” model. Nonparametric models, unlike geometrical models such as NNs and polynomial iterations, require raw data to make predictions. The kernel hyperparameters are GP-optimized [41].

### 2.3. M5P Tree Model

Quinlan (1992) [42] developed M5P algorithm model trees which efficiently handle large datasets with many dimensions and attributes. Missing data will not create ambiguity. This tree algorithm applies multivariate linear regression at each branching node. Model trees are two-stage. A splitting criterion generates a decision tree. The M5P tree model splits based on predicted error reduction from evaluating each characteristic at a network and error quantization from managed data instances entering a node. After expanding every result, it determines which attribute is the lowest in the normal [43]. This method employs standard deviation to measure terminal node error and creates linear functions at each node, purifying the data. The standard deviation reduction (*SDR*) formula is:(1)$$SDR=sd\left(Y\right)-\frac{\sum_{i=1}^{x}\left|Y_{i}\right|}{\left|Y\right|}*sd\left(Y\right).$$
where *Y* = number of samples; *Y_i_ =* number of samples representing *i*th sample having potential rise; and *sd* = standard deviation.

### 2.4. RT Model

RT node is a tree-based classification and regression method. Bagged decision trees are created using random data. Each tree node uses the best variable split. Random forest separates nodes by the greatest random predictor. Random trees sample using replacement and bootstrap. Sample data generates a tree model. Random trees never resample. Instead, it randomly picks a subset of predictors to divide a tree node. For each tree node, repeat the technique. Random tree growth works like this. Random tree models work well with big data and numerous fields. Bagging and field samples prevent overfitting, making test findings more repeatable (Kalmegh 2015) [44,45].

### 2.5. MLR Model

Multiple linear regression is one modeling technique used to explain the effect of influential variables used independently of one another [46]. Generally, the MLR model can be expressed as in Equation (1):P = q2 + c1q1 + c2q2 + c3q3 + … + cnqn(2)
where P = dependent variable;q1 … qn = independent variable;q2 = Regression Coefficient.

Parameter values were estimated using least-squares techniques. The best MLR takes into account a variety of statistical criteria, such as the smallest RSME, the highest correlation, the largest F statistic, and the largest number of descriptors [47].

## 3. Methodology

The materials used to conduct the experiments included bitumen, glass fiber, carbon fiber, and filler, as well as open-graded coarse aggregates. The detailed methodology of the experiment performed is shown in Figure 2. Specific requirements of the material as shown in Section 3.1, Section 3.2 and Section 3.3.

### 3.1. Aggregates

In this study, a 20-mm coarse aggregate size is used to produce asphalt mix. Table 2 and Table 3 depict the coarse aggregate (CA) and fine aggregate (FA) grading as per (ASTM D-6913:04) [48] and Table 4 summarizes the physical properties of the aforementioned aggregates [49,50,51,52].

### 3.2. Bitumen

A PG of (80–100) bitumen was utilized for this study which was sourced from the HPPWD in Solan, India. Table 5 lists bitumen’s basic characteristics [53,54,55,56].

### 3.3. Glass and Carbon Fibers

Chopped glass and carbon fiber were the two types of fibers utilized. Five different types of asphalt mixes, including GF, CF, and glass and carbon fiber hybrid mixes, were prepared. Table 6 summarizes the properties of glass and carbon fibers.

## 4. Experimental Investigation

The asphalt mix was developed following the specifications specified by ASTM D-1559 [57]. Cylindrical specimens with a diameter of 101.6 mm × 63.5 mm in height were used. A total of 1200 gm of open-graded coarse aggregate was utilized and thoroughly oven dried at a temperature between 100–110 °C for 24 h. The aggregate was heated at a temperature of 170 °C to 190 °C and blended with asphalt at 160 °C. In both the control mix and glass- and carbon-fiber-modified asphalt mixtures, the percentages of glass fiber and carbon content that were chosen were 0%, 1.0%, 2.0%, 3.0%, and 4.0%, and the asphalt content varied from 4.5 to 6.0% at 0.5% intervals, respectively. After the mixture was placed into the mould, it was compacted with 75 blows on both sides with 4.5 kg sliding weight after the compacting sample was extracted using a sample extractor. The design mix of glass and carbon is depicted in Table 7. Figure 3a–c shows the samples were made using glass fiber, carbon fiber, 75GF:25CF, 50GF:50CF, and 25GF:75CF hybrid asphalt mix. The Marshall stability testing apparatus as well as the testing of the Marshall specimen are depicted in Figure 4a,b.

### Collection of Dataset

For the Marshall stability prediction, a total of 164 observations are incorporated by using experimental data of glass and carbon fibers and variations in both fibers provided in Table 8. After that, the total observations were split, at random, into two different subsets, each of which contained a 70/30 ratio having 110 observations in the training and 54 in the testing dataset, respectively. Table 9 provides a summary of the data sets obtained from the experiments. For the prediction of MS, five types of ML techniques (i.e., ANN, GP, M5P Tree, RT, and MLR) were implemented using Weka 3.9.5 software and ten input parameters including (BC), (GF), (CF), 75GF:25CF, 50GF:50CF, 25GF:75CF, (VG), (FL), and (FD) glass and (FD) carbon, respectively, were assessed. The statistical characteristics of said input parameters are shown in Table 10. The input characteristics were evaluated to predict the outcome, i.e., Marshall stability of hybrid asphalt concrete, using the performance evaluation parameters that are illustrated in Section 5.

## 5. Performance Evaluating Parameters

The effectiveness of each model was judged with reference to the following five statistical metrics: CC, which can vary from −1 to 1 (higher correlation coefficients indicate more accurate findings), MAE, RMSE, RAE, and RRSE. The RMSE and MAE are two forms of error that represent the average deviation between actual and predicted values. The better the prediction, the lower the error. These statistics measure the difference between actual and predicted results for the same behavior, i.e., a smaller computed error indicates improved output outcomes. This may be determined using the formula stated in Equations (3)–(7) below:(3)$$\mathrm{CC}=\frac{\sum_{i=1}^{n}\left(L_{i}-\overset{\_}{L}\right)\left(G_{i}-\overset{\_}{G}\right)}{\sqrt{\sum_{i=1}^{n}{\left(L-\overset{\_}{L}\right)}^{2}}\sqrt{\sum_{i=1}^{n}{\left(G_{i}-\overset{\_}{G}\right)}^{2}}}.$$
(4)$$\mathrm{MAE}=\frac{1}{n}(\sum_{i=1}^{n}\left|L-G\right|).$$
(5)$$\mathrm{RMSE}=\surd \frac{1}{n}\sum_{i=1}^{n}{\left(L-G\right)}^{2}.$$
(6)$$\mathrm{RAE}=\frac{\sum_{i=1}^{n}\left|L-G\right|}{\sum_{i=1}^{n}\left(\left|L-\bar{L}\right|\right)}.$$
(7)$$\mathrm{RRSE}=\sqrt{\;\frac{\sum_{i=1}^{n}{\left(L-G\right)}^{2}}{\sum_{i=1}^{n}{\left(\left|L-\bar{G}\right|\right)}^{2}}}$$
where *L* = actual values; *G* = average observation; $\bar{G}$ = predicted value; and *n* = number of observations.

## 6. Results and Discussion

After obtaining the 164 observations from the various experimental work, the total data set was generated for prediction and analyzing the performance of five types of asphalt mixes, i.e., glass fiber asphalt mix, carbon fiber asphalt mix, and three glass-carbon fiber (25:75, 50:50, 75:25 proportions) combination asphalt mixes for Marshall stability. The performance of such mixes can be assessed by analyzing each applied model and is discussed in the following section.

### 6.1. ANN Model Performance Assessment

A multilayer perceptron model serves as the core of the iterative process that constitutes ANN-based model generation. Several efforts were made to find the ideal value with the maximum defined CC value with the fewest errors for training and testing the dataset for assessing the generated models’ predictions. The user-defined parameters that were utilized in the process of evaluating the ANN model included the sigmoid activation function node (1–9), learning rate (0.2), momentum (0.1), number of iterations (1700), hidden layer (1), and number of neurons (20) [58,59,60,61,62,63]. Table 12 shows the performance comparison of ANN and MLR model which depicts that an ANN-based model outperforms other models for predicting the MS of modified AC for training and testing stages, with the value of CC as (0.8858, 0.8392), R^2^ (0.7846, 0.7042), MAE as (1.4449, 1.4996), RMSE as (1.8391, 1.8315), RAE as (58.25%, 63.07%) and RRSE as (58.44%, 58.89%), respectively. Figure 5a,b represents the training and testing stages; this indicates that the majority of the scattered data points fall inside and lie within perfect line agreement, which shows an ideal match between actual and predicted values and also falls within the ±20% error range.

### 6.2. GP Model Performance Assessment

In Gaussian processes, a regression technique with parameters such as (O = 2.0 and S = 2.0), noise = (1.0), and seed = (1.0) is used in conjunction with a universal kernel (PUK) based on the Pearson VII function. According to the findings presented in Table 11, a GP-PUK-based model appears to be reliable for predicting the MS of modified asphalt concrete, with values of CC as (0.8383, 0.8187), R^2^ as (0.7027, 0.6702), MAE as (1.4276, 1.5350), RMSE as (1.7688, 1.8524), RAE as (57.55%, 64.56%), and RRSE as (56.21%, 59.57%) for both stages. Figure 6a,b shows the agreement line that connects the actual and the predicted values in which it can be seen from the scatter points that most of the predicted values fall within the ±25% error range [64,65,66,67].

### 6.3. M5P Model Performance Assessment

The performance assessment of the M5P model was evaluated using a pruned model tree (using smoothed linear models). The outcome of Table 11 depicts that the M5P tree model is consistent in predicting the MS of modified AC with the value of CC as (0.8396, 0.8172), R^2^ as (0.7049, 0.6678), MAE as (1.3358, 1.5264), RMSE as (1.7138, 1.8331), RAE as (53.85%, 64.20%), and RRSE as (54.46%, 58.94%) for both stages. Figure 7a,b presents an agreement graph that plots actual and predicted values and shows most of the scatter data points lie closer to the agreement line using M5P tree-based models. The graph displays that the predicted values fall within the margin of error of ±25% at both phases [68,69,70,71,72].

### 6.4. RT Model Performance Assessment

The performance of a random tree is based on the decision tree and class for constructing a tree that considers K randomly chosen attributes at each node, i.e., value of K = 8, number of folds = 4, and number of seed = 8. The performance assessment of the RT model depicted in Table 11 indicates that the RT model is quite competitive with other models in predicting the MS of modified AC, with the value of CC as (0.8414, 0.7936), R^2^ as (0.7079, 0.6298), MAE as (1.2008, 1.6573), RMSE as (0.0171, 1.9848), RAE as (48.41%, 69.70%), and RRSE as (54.42%, 63.80%) for both stages, respectively. The agreement graph between the actual value and the predicted value is shown in Figure 8a,b; it shows that most of the data points are relatively near to the actual values in both the training and testing phases which fall within the margin error of ±28% in training and ±30% in the testing stage [73,74,75].

### 6.5. MLR Model Performance Assessment

The MLR analysis was performed and it can be seen from Table 12 that the performance of MLR shows overfitting of datasets in training and testing stages for the prediction of MS, with CC as (0.7647, 0.7976), R^2^ as (0.5847, 0.6361), MAE as (1.6509, 1.6387), RMSE as (2.0278, 1.8910), RAE as (66.55%, 68.92%), and RRSE as (64.44%, 60.81%) for both stages. Figure 9a,b shows that the majority of the predicted data points are scattered which falls within the margin of error of ±30% in training and ±25% in the testing stage.

The following equation, which indicates the sign and the magnitude of each feature’s contribution to the modelled asphalt property, is obtained by using the MLR model as given in Equation (8).
MS (kN) = 1.2356 × Bitumen content (%) + (−0.4678) × 75GF:25CF(%) + (−0.5668) × 25GF:75CF (%) + (1.1838) × Fiber Diameter (Carbon) + 3.4081(8)

The impact of the hybrid mix 50GF-50CF (%), GF (%), CF (%), FD (glass), type of the bitumen (VG), and fiber length (FL) on the Marshall stability is found to be negligible due to their constant values. Hence, they did not figure in the MLR model equation.

## 7. Comparison of Machine Learning Models

The MS predictions of AC incorporating glass and carbon fibers, as well as variations in both fibers with the ratios 75GF:25CF, 50GF:50CF, and 25GF:75CF, were examined in this study by implementing five ML techniques. Ten attributes including BC, GF, CF, 75GF:25CF, 50GF:50CF, and 25GF:75CF, VG, (FL), and (FD) glass, and (FD) carbon, as well as Marshall stability (MS) as an output parameter and Equations (2)–(6) were used to evaluate the input parameters. Table 12 represents the comparison of the ANN model is applied with least performing model for training and testing stages, which suggests that the ANN-based model has outperformed the other, with CC as (0.8858, 0.8392), R^2^ (0.7846, 0.7042), MAE as (1.4449, 1.4996), RMSE as (1.8391, 1.8315), RAE as (58.25% 63.07%), and RRSE as (58.44%, 58.89%), respectively. Figure 10a,b displays the results of the performance of all the models used in both stages, showing that all models’ prediction values are very near to the actual data, with a ±30 error bandwidth in the training and testing stage. The median and quartile values of actual and predicted MS are shown in Table 13, indicating the representation of data central tendency as a function of the first five numbers and depicts the highest predicted model has an IQR of 4.029, which shows the range of scores from the lower to upper quartile. The data distribution for each model is shown in Figure 11 as a boxplot with percentile labels and the red symbol ’+’ shows outlier point. This plot demonstrates that the ANN model uses more accurate techniques of data distribution, and hence outperforms in predicting the MS of the modified asphalt mix. Predicted Marshall stability and relative error with data set numbers for all training and testing models are shown in Figure 12a,b, indicating that the ANN model has fewer error bands that are within the range of statistical significance (−3 to 3).

The results indicated that the artificial neural network outperformed other models in predicting the Marshall stability of modified asphalt concrete with a higher value of the coefficient of correlation (0.8392), R^2^ (0.7042), and a lower mean absolute error value (1.4449) and root mean square error value (1.8315) in the testing stage with a small error band; furthermore, it provided the best optimal fit for predicting the output. The results of the sensitivity analysis show that the carbon-fiber asphalt mix is the most effective parameter, followed by glass-carbon fiber (50:50 proportion) modified asphalt mix, which influences the Marshall strength to a greater extent. The results obtained from the sensitivity analysis performed with the ANN model showed that the carbon-fiber asphalt mix was the most sensitive to Marshall stability among all the five applied asphalt mixes. In the prior study done by [76], results from the research demonstrated that the ANN technique performed better than regression models for predicting rutting performance using carbon nanotubes. The analysis further showed that the glass-fiber asphalt mix is the weakest among all the mixes.

## 8. Feature Importance

The feature importance analysis was performed with the MLR model, as shown in Table 14, to determine the sensitivity of each parameter to MS of the modified asphalt mixes as the slight nonlinearity of the problem identified by the NN model being slightly better than the MLR model, making feature importance complex. The purple box in each row represents the impact on the shown indices in the table by non-consideration of the boxed input parameter in the corresponding column, whereas row 1 (without box) represents the consideration of all input parameters in the feature importance analysis. The results of the analysis show that the first five input variables are the top fifth most sensitive parameters in both models. The carbon diameter in asphalt mix, followed by bitumen content, has been proven to be the most sensitive material, having a lower coefficient of correlation with a higher magnitude of errors. Therefore, the first five input variables, i.e., carbon fiber diameter, bitumen content, hybrid asphalt mix of glass-carbon fiber in 75:25 percent, carbon fiber content, and hybrid asphalt mix of glass-carbon fiber in 50:50 percent, are highly sensitive parameters that influence the Marshall strength of the modified asphalt mixes to a greater extent.

## 9. Conclusions

The current study examined the Marshall stability of five types of modified asphalt mixes blended with glass, carbon, and glass-carbon fibers using five machine learning techniques, namely ANN, GP-PUK, M5P, RT, and MLR-based models. The performance evaluation results revealed that the artificial neural network (ANN) outperformed the other models in predicting the Marshall stability of modified asphalt mix, with the CC as 0.8392, R^2^ as 0.7042, MAE as 1.4996, RMSE as 1.8315, RAE as 63.07%, and RRSE as 58.89% for the testing dataset. An agreement graph showed that ANN had a smaller error band and optimal fit for predicting the Marshall stability. The results of the feature importance analysis indicate that the first five input variables, i.e., carbon fiber diameter, bitumen content, hybrid asphalt mix of glass-carbon fiber in 75:25 percent, carbon fiber content, and hybrid asphalt mix of glass-carbon fiber in 50:50 percent, are highly sensitive parameters which influence the Marshall strength of the modified asphalt mixes to a greater extent. Five types of asphalt mixes, i.e., glass-fiber asphalt mix, carbon-fiber asphalt mix, and three glass-carbon fiber (25:75, 50:50, 75:25 proportions) combination asphalt mixes were utilized in this investigation. The interval of the proportion of the glass-fiber combination in the modified asphalt mix can be shortened for precise results. Furthermore, the machine learning algorithm can be explored for Marshall stability predictions vis-à-vis the sensitivity analysis.

## Figures and Tables

**Figure 1 materials-15-08944-f001:**
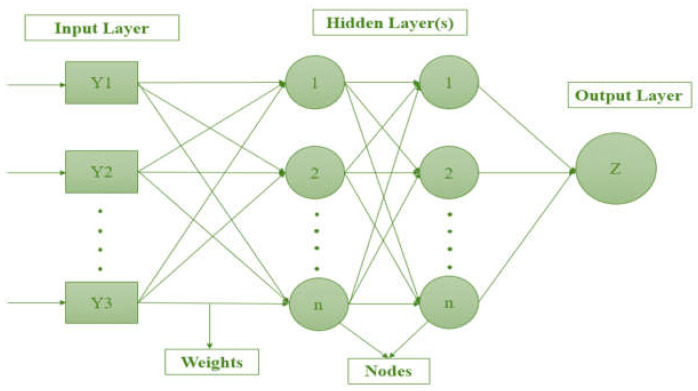
ANN multilayer perceptron structure.

**Figure 2 materials-15-08944-f002:**
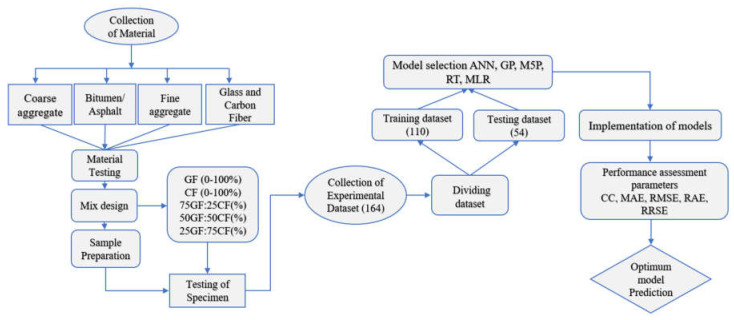
Flowchart showing the detailed methodology of the present study.

**Figure 3 materials-15-08944-f003:**
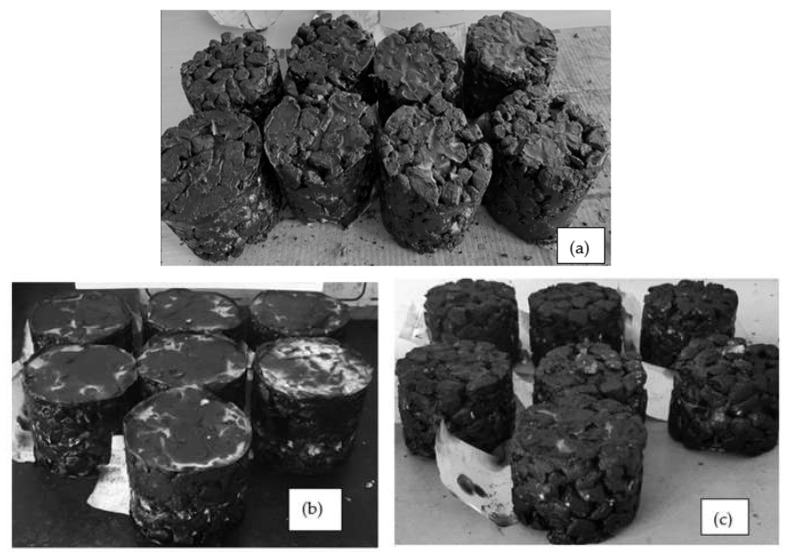
(**a**–**c**) Glass and carbon fiber with variation in fiber specimen ranging from 0–4.0%.

**Figure 4 materials-15-08944-f004:**
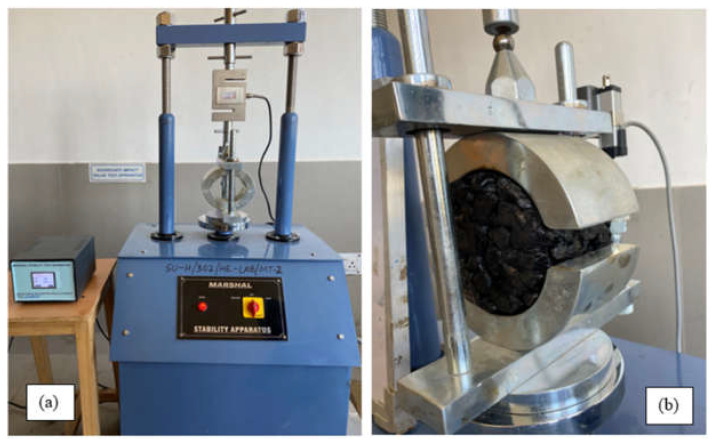
(**a**,**b**) Marshall stability testing apparatus and testing of the Marshall specimen.

**Figure 5 materials-15-08944-f005:**
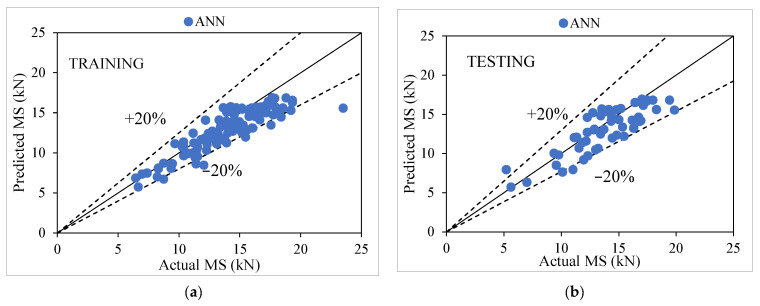
(**a**,**b**). Agreement graph showing actual vs. predicted values of MS using an ANN-based model for both stages.

**Figure 6 materials-15-08944-f006:**
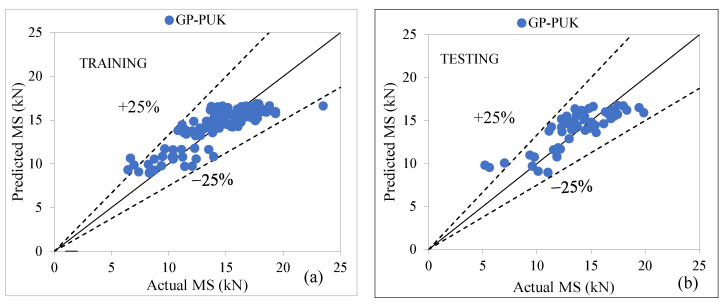
(**a**,**b**) Agreement graph showing actual vs. predicted values of MS by using a GP-PUK-based model for both stages.

**Figure 7 materials-15-08944-f007:**
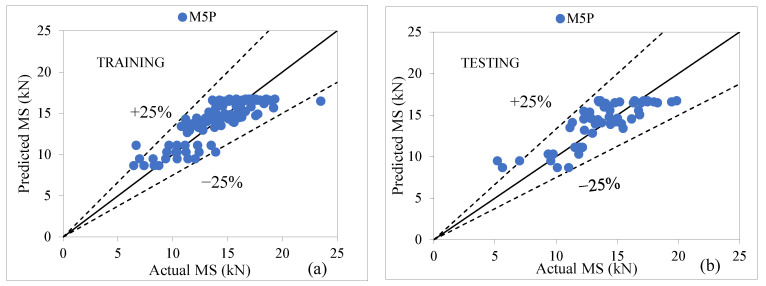
(**a**,**b**) Agreement graph showing actual vs. predicted values of MS by using an M5P tree-based model for both stages.

**Figure 8 materials-15-08944-f008:**
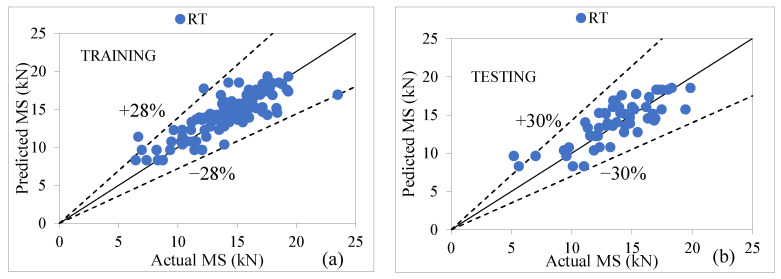
(**a**,**b**). Agreement graph showing actual vs. predicted values of MS by using an RT-based model for both stages.

**Figure 9 materials-15-08944-f009:**
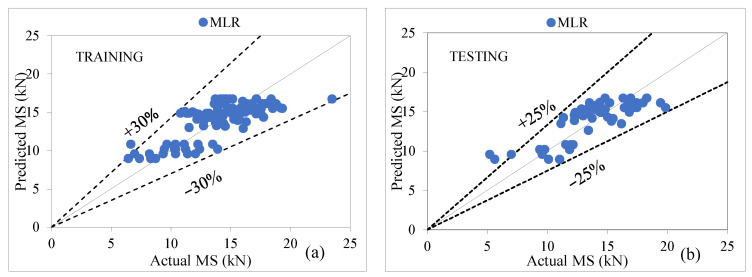
(**a**,**b**) Agreement graph showing actual vs. predicted values of MS by using an MLR model for both stages.

**Figure 10 materials-15-08944-f010:**
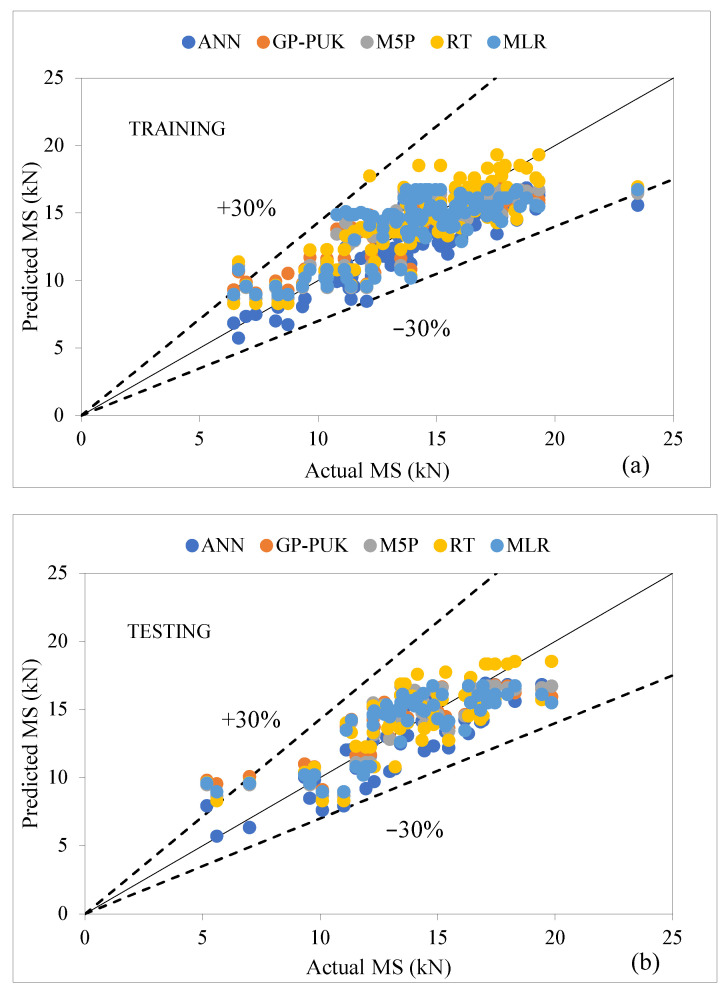
(**a**,**b**) Comparison of all models using training and testing stages.

**Figure 11 materials-15-08944-f011:**
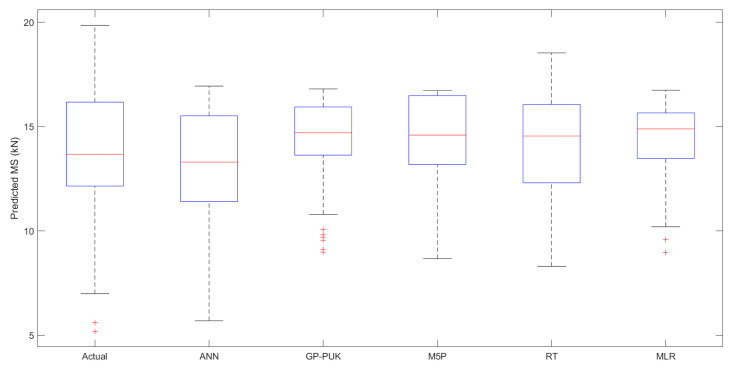
Boxplot with all applied models using the testing stage.

**Figure 12 materials-15-08944-f012:**
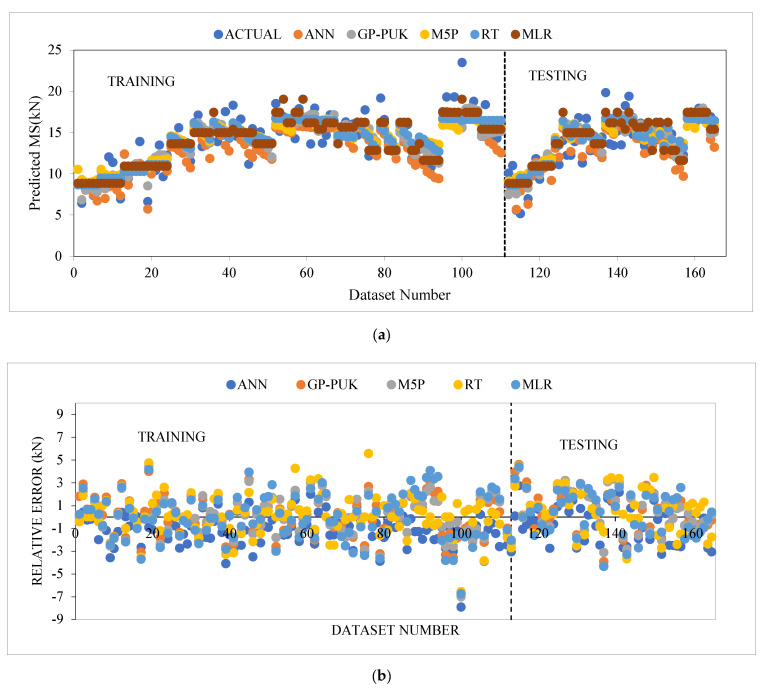
(**a**) Predicted MS of all models applied for training and testing stages. (**b**) Relative error with dataset using all models for training and testing stages.

**Table 1 materials-15-08944-t001:** Comparison of the approaches used by previous authors.

Sr. No.	Authors	Additive Used	Technique Applied	Output	Findings
1.	Vadood et al., [30]	Modified HMA samples using polypropylene and polyester fibers (hybrid and single modes)	Artificial neural network; genetic algorithm	Resilient modulus of the modified Hot Mix Asphalt	ANN with two neurons per layer can accurately predict fiber-reinforced HMA’s resilient modulus.
2.	Karahancer et al., [31]	Polyparaphenylene Terephtalamide fiber (PTF) rate	ANN	Predicting Marshall stability of asphalt pavement	With a regression value of 96%, the ANN model accurately predicted the experimental parameters.
3.	Awan et al., [32]	-	Multi-Expression Programming (MEP);	Marshall Stability (MS) and Marshall Flow (MF) for Asphalt Base Course (ABC) and Asphalt Wearing Course (AWC) of flexible pavements.	The developed models have generated outcomes that are in agreement with the experimental data. Function and data work equally well for unknowable data.
4.	Ameri et al., [5]	Glass and basalt fiber	ANFIS	Indirect tensile strength, moisture sensitivity, resilient modulus, and creep tests using the Marshall test	The developed ANFIS models are capable of predicting output values that are close to actual data.
5.	Hejazi et al., [33]	Glass, nylon 6.6, polypropylene, and polyester	ANN	Marshall test results in terms of stability, flow, and specific gravity	The models concluded that glass, polyester, and nylon were better, and they were suggested for predicting any textile fibers that may be used in AC.
6.	Mirabdolazimi and Shafabakhsh [34]	Forta fiber	Artificial neural networks, Genetic programming	Assess the rutting resistance of asphalt samples	(ANN) model for rutting depth showed good agreement with experimental results, whereas the genetic programming model is very effective.
7.	Yardim et al., [35]	Hydrated lime	Fuzzy logic, artificial neural networks.	Marshall design test parameters of hot mix asphalt samples	The developed models provided reasonable estimations of the mixture parameters.
8.	Olukanni et al., [36]	-	MLR and Genetic Programming Method.	Determine Marshall test outcomes including stability, flow, and Marshall quotient	The GP models outperform the MLR models in terms of R^2^ and lower error.
9.	Xiao et al., [37]	Rubberized Asphalt Concrete	Neural network	Determine the ultimate fatigue life of the modified mixtures.	ANN techniques are superior to the conventional statistical prediction model for predicting the fatigue life of the modified mixtures tested.
10.	Babagoli and Rezaei [38]	*Styrene-butadiene rubber*, Crumb Rubber	Artificial neural networks (ANN) Support vector regression (SVR)	The fracture energy (FE), indirect tensile strength (ITS), and resilient modulus (Mr) of mixtures	The outcomes demonstrated that ANN consistently outperformed SVR.

**Table 2 materials-15-08944-t002:** Coarse aggregate grading.

Sieve Size (mm)	25	20	16	12.5	10	4.75
Passing (%)	100	97.67	67.47	30.07	8.27	0

**Table 3 materials-15-08944-t003:** Fine aggregate grading.

Sieve Size (mm/mic)	10	4.75	2.36	1.18	600	300	150	7
Passing (%)	98.4	93.6	89.8	86.0	76.9	19.9	7.4	4.8

**Table 4 materials-15-08944-t004:** Physical characteristics of the coarse and fine aggregates.

Test Properties	CA	FA	Standard Specifications
SG	2.63	2.42	ASTM C-128 [49]
Apparent SG (gm/cm^3^)	2.83	2.47	
Water Absorption (%)	2.75	0.33	
Bulk SG (gm/cm^3^)	1.51	1.68	
Crushing Value Test (%)	23.43	–	ASTM C-127 [50]
Impact Value Test (%)	*7.95*	–	
LAV Test (%)	34.34	–	ASTM C-131 [51]
(FI) and (EI) index (%)	14.64, 8.64	–	ASTM D- 4791 [52]

**Table 5 materials-15-08944-t005:** Bitumen characteristics.

Test on Bitumen	Standard Specifications	Value
SG (25 °C)	ASTM-D70 [53]	0.99
Penetration 25 °C, (0.1 mm)	ASTM-D5 [54]	97.66
Flash Point °C	ASTM-D92 [55]	281
Softening Point Test °C	ASTM-D36 [56]	39.2

**Table 6 materials-15-08944-t006:** Properties of glass and carbon fibers.

Properties of Fibers	GF	CF
Length (mm)	12	12
Diameter (µm)	15	5
Color	White	Black
Tensile strength (Mpa)	4700–4800	5790
Elongation (%)	5.7	-
Density (gm/cc)	2.46	1.80
Failure strain (%)	-	2.0
Base	S-glass	PAN-fiber

**Table 7 materials-15-08944-t007:** Design mix of glass and carbon fibers.

GF (%)	GF:CF (%)	GF:CF (%)	GF:CF (%)	CF (%)
100:0	75:25	50:50	25:75	0:100

**Table 8 materials-15-08944-t008:** Experimental dataset.

No. of Specimen	Glass Fiber (100) %	75GF:25CF (%)	50GF:50CF (%)	25GF:75CF (%)	Carbon Fiber (100) %	Bitumen Grade	Fiber Length (mm)	Fiber Diameter (Glass)	Fiber Diameter (Carbon)	Marshall Stability (kN)
1	0	0	0	0	0	10	0	0	0	8.73
2	0.5	0	0	0	0	10	12	15	0	6.44
3	1	0	0	0	0	10	12	15	0	10.1
4	1.5	0	0	0	0	10	12	15	0	8.31
5	2	0	0	0	0	10	12	15	0	8.31
6	2.5	0	0	0	0	10	12	15	0	11.01
7	3	0	0	0	0	10	12	15	0	7.37
8	3.5	0	0	0	0	10	12	15	0	8.73
9	4	0	0	0	0	10	12	15	0	5.61
10	0	0	0	0	0	10	0	0	0	10.39
11	0.5	0	0	0	0	10	12	15	0	8.206
12	1	0	0	0	0	10	12	15	0	5.19
13	1.5	0	0	0	0	10	12	15	0	12.05
14	2	0	0	0	0	10	12	15	0	11.4
15	2.5	0	0	0	0	10	12	15	0	9.56
16	3	0	0	0	0	10	12	15	0	9.35
17	3.5	0	0	0	0	10	12	15	0	6.96
18	4	0	0	0	0	10	12	15	0	7
19	0	0	0	0	0	10	0	0	0	12.4
20	0.5	0	0	0	0	10	12	15	0	9.45
21	1	0	0	0	0	10	12	15	0	9.76
22	1.5	0	0	0	0	10	12	15	0	10.29
23	2	0	0	0	0	10	12	15	0	11.22
24	2.5	0	0	0	0	10	12	15	0	11.84
25	3	0	0	0	0	10	12	15	0	13.92
26	3.5	0	0	0	0	10	12	15	0	10.39
27	4	0	0	0	0	10	12	15	0	9.35
28	0	0	0	0	0	10	0	0	0	6.65
29	0.5	0	0	0	0	10	12	15	0	11.11
30	1	0	0	0	0	10	12	15	0	11.53
31	1.5	0	0	0	0	10	12	15	0	10.38
32	2	0	0	0	0	10	12	15	0	13.5
33	2.5	0	0	0	0	10	12	15	0	12.15
34	3	0	0	0	0	10	12	15	0	9.66
35	3.5	0	0	0	0	10	12	15	0	12.26
36	4	0	0	0	0	10	12	15	0	11.95
37	0	0.5	0	0	0	10	12	15	5	13.40
38	0	1	0	0	0	10	12	15	5	14.03
39	0	1.5	0	0	0	10	12	15	5	13.72
40	0	2	0	0	0	10	12	15	5	15.59
41	0	2.5	0	0	0	10	12	15	5	14.03
42	0	3	0	0	0	10	12	15	5	11.12
43	0	3.5	0	0	0	10	12	15	5	13.82
44	0	4	0	0	0	10	12	15	5	11.53
45	0	0.5	0	0	0	10	12	15	5	14.03
46	0	1	0	0	0	10	12	15	5	16.21
47	0	1.5	0	0	0	10	12	15	5	17.14
48	0	2	0	0	0	10	12	15	5	12.26
49	0	2.5	0	0	0	10	12	15	5	13.30
50	0	3	0	0	0	10	12	15	5	15.27
51	0	3.5	0	0	0	10	12	15	5	12.26
52	0	4	0	0	0	10	12	15	5	13.74
53	0	0.5	0	0	0	10	12	15	5	16.00
54	0	1	0	0	0	10	12	15	5	13.92
55	0	1.5	0	0	0	10	12	15	5	13.92
56	0	2	0	0	0	10	12	15	5	16.00
57	0	2.5	0	0	0	10	12	15	5	16.83
58	0	3	0	0	0	10	12	15	5	17.56
59	0	3.5	0	0	0	10	12	15	5	14.34
60	0	4	0	0	0	10	12	15	5	11.33
61	0	0.5	0	0	0	10	12	15	5	18.32
62	0	1	0	0	0	10	12	15	5	14.55
63	0	1.5	0	0	0	10	12	15	5	16.73
64	0	2	0	0	0	10	12	15	5	16.63
65	0	2.5	0	0	0	10	12	15	5	14.75
66	0	3	0	0	0	10	12	15	5	12.83
67	0	3.5	0	0	0	10	12	15	5	11.17
68	0	4	0	0	0	10	12	15	5	15.48
69	0	0	0.5	0	0	10	12	15	5	12.88
70	0	0	1	0	0	10	12	15	5	14.26
71	0	0	1.5	0	0	10	12	15	5	15.08
72	0	0	2	0	0	10	12	15	5	13.40
73	0	0	2.5	0	0	10	12	15	5	13.20
74	0	0	3	0	0	10	12	15	5	13.10
75	0	0	3.5	0	0	10	12	15	5	14.44
76	0	0	4	0	0	10	12	15	5	12.05
77	0	0	0.5	0	0	10	12	15	5	18.53
78	0	0	1	0	0	10	12	15	5	19.85
79	0	0	1.5	0	0	10	12	15	5	15.17
80	0	0	2	0	0	10	12	15	5	16.94
81	0	0	2.5	0	0	10	12	15	5	13.62
82	0	0	3	0	0	10	12	15	5	16.64
83	0	0	3.5	0	0	10	12	15	5	15.80
84	0	0	4	0	0	10	12	15	5	13.44
85	0	0	0.5	0	0	10	12	15	5	14.26
86	0	0	1	0	0	10	12	15	5	17.90
87	0	0	1.5	0	0	10	12	15	5	15.19
88	0	0	2	0	0	10	12	15	5	16.74
89	0	0	2.5	0	0	10	12	15	5	16.35
90	0	0	3	0	0	10	12	15	5	13.51
91	0	0	3.5	0	0	10	12	15	5	13.63
92	0	0	4	0	0	10	12	15	5	17.98
93	0	0	0.5	0	0	10	12	15	5	18.29
94	0	0	1	0	0	10	12	15	5	15.17
95	0	0	1.5	0	0	10	12	15	5	13.73
96	0	0	2	0	0	10	12	15	5	23.50
97	0	0	2.5	0	0	10	12	15	5	14.69
98	0	0	3	0	0	10	12	15	5	17.17
99	0	0	3.5	0	0	10	12	15	5	14.80
100	0	0	4	0	0	10	12	15	5	17.16
101	0	0	0	0.5	0	10	12	15	5	14.39
102	0	0	0	1	0	10	12	15	5	15.01
103	0	0	0	1.5	0	10	12	15	5	15.75
104	0	0	0	2	0	10	12	15	5	16.30
105	0	0	0	2.5	0	10	12	15	5	16.17
106	0	0	0	3	0	10	12	15	5	14.54
107	0	0	0	3.5	0	10	12	15	5	16.06
108	0	0	0	4	0	10	12	15	5	13.42
109	0	0	0	0.5	0	10	12	15	5	14.71
110	0	0	0	1	0	10	12	15	5	15.81
111	0	0	0	1.5	0	10	12	15	5	12.74
112	0	0	0	2	0	10	12	15	5	17.75
113	0	0	0	2.5	0	10	12	15	5	12.18
114	0	0	0	3	0	10	12	15	5	15.34
115	0	0	0	3.5	0	10	12	15	5	14.42
116	0	0	0	4	0	10	12	15	5	12.74
117	0	0	0	0.5	0	10	12	15	5	14.14
118	0	0	0	1	0	10	12	15	5	19.20
119	0	0	0	1.5	0	10	12	15	5	16.60
120	0	0	0	2	0	10	12	15	5	14.34
121	0	0	0	2.5	0	10	12	15	5	13.90
122	0	0	0	3	0	10	12	15	5	12.74
123	0	0	0	3.5	0	10	12	15	5	15.47
124	0	0	0	4	0	10	12	15	5	12.75
125	0	0	0	0.5	0	10	12	15	5	16.01
126	0	0	0	1	0	10	12	15	5	14.39
127	0	0	0	1.5	0	10	12	15	5	13.89
128	0	0	0	2	0	10	12	15	5	16.01
129	0	0	0	2.5	0	10	12	15	5	14.83
130	0	0	0	3	0	10	12	15	5	11.82
131	0	0	0	3.5	0	10	12	15	5	12.33
132	0	0	0	4	0	10	12	15	5	12.97
133	0	0	0	0	0.5	10	12	0	5	12.98
134	0	0	0	0	1	10	12	0	5	13.09
135	0	0	0	0	1.5	10	12	0	5	13.2
136	0	0	0	0	2	10	12	0	5	11.43
137	0	0	0	0	2.5	10	12	0	5	10.8
138	0	0	0	0	3	10	12	0	5	12.31
139	0	0	0	0	3.5	10	12	0	5	11.53
140	0	0	0	0	4	10	12	0	5	11.32
141	0	0	0	0	0.5	10	12	0	5	17.45
142	0	0	0	0	1	10	12	0	5	17.56
143	0	0	0	0	1.5	10	12	0	5	19.32
144	0	0	0	0	2	10	12	0	5	16.41
145	0	0	0	0	2.5	10	12	0	5	17.35
146	0	0	0	0	3	10	12	0	5	19.32
147	0	0	0	0	3.5	10	12	0	5	17.14
148	0	0	0	0	4	10	12	0	5	17.16
149	0	0	0	0	0.5	10	12	0	5	19.43
150	0	0	0	0	1	10	12	0	5	17.97
151	0	0	0	0	1.5	10	12	0	5	17.87
152	0	0	0	0	2	10	12	0	5	17.66
153	0	0	0	0	2.5	10	12	0	5	17.45
154	0	0	0	0	3	10	12	0	5	18.8
155	0	0	0	0	3.5	10	12	0	5	17.66
156	0	0	0	0	4	10	12	0	5	17.03
157	0	0	0	0	0.5	10	12	0	5	14.54
158	0	0	0	0	1	10	12	0	5	18.39
159	0	0	0	0	1.5	10	12	0	5	16.93
160	0	0	0	0	2	10	12	0	5	14.96
161	0	0	0	0	2.5	10	12	0	5	14.13
162	0	0	0	0	3	10	12	0	5	16.31
163	0	0	0	0	3.5	10	12	0	5	14.34
164	0	0	0	0	4	10	12	0	5	15.17

**Table 9 materials-15-08944-t009:** Details of the experimental dataset.

S. No.	BC (%)	GF (%)	75GF:25CF	50GF:50CF	25GF:75CF	CF (%)	(VG)	FL (mm)	FD Glass Fiber (µm)	FD Carbon Fiber (µm)	MS (kN)	No. of Observations from Current Research
**Dataset Range**												
1.	4.5–6.0	0–4.0	-	-	-	-	10	12	15	5	5.19–13.92	36
2.	4.5–6.0	-	-	-	-	0.5–4.0	10	12	15	5	12.31–19.32	32
3.	4.5–6.0	-	0.5–4.0	-	-	-	10	12	15	5	11.12–18.32	32
4.	4.5–6.0	-	-	0.5–4.0	-	-	10	12	15	5	12.05–23.50	32
5.	4.5–6.0	-	-		0.5–4.0	-	10	12	15	5	11.82–19.20	32
Total observations												164

**Table 10 materials-15-08944-t010:** Statistical characteristics of the dataset.

**Training**											
	**BC (%)**	**GF (%)**	**75GF:25CF**	**50GF:50CF**	**25GF:75CF**	**CF (%)**	**(VG)**	**FL(mm)**	**FD Glass Fiber (µm)**	**FD Carbon Fiber (µm)**	**MS (kN)**
Mean	5.2545	0.3818	0.4500	0.4545	0.4227	0.4455	10.0000	11.5636	11.5909	3.9091	14.0780
Standard Error	0.0543	0.0890	0.0998	0.0991	0.0958	0.0999	0.0000	0.2152	0.6021	0.1978	0.3014
Median	5.2500	0.0000	0.0000	0.0000	0.0000	0.0000	10.0000	12.0000	15.0000	5.0000	14.3
Standard Deviation	0.5697	0.9334	1.0469	1.0395	1.0050	1.0478	0.0000	2.2566	6.3148	2.0745	3.1614
Standard Variance	0.3245	0.8712	1.0961	1.0805	1.0100	1.0979	0.0000	5.0922	39.8770	4.3036	9.994
Range	1.5000	3.5000	4.0000	4.0000	4.0000	4.0000	0.0000	12.0000	15.0000	5.0000	17.06
Minimum	4.5000	0.0000	0.0000	0.0000	0.0000	0.0000	10.0000	0.0000	0.0000	0.0000	6.44
Maximum	6.0000	3.5000	4.0000	4.0000	4.0000	4.0000	10.0000	12.0000	15.0000	5.0000	23.5
Confidence Level (95.0%)	0.1077	0.1764	0.1978	0.1964	0.1899	0.1980	0.0000	0.4264	1.1933	0.3920	0.59742
**Testing**											
	**BC (%)**	**GF (%)**	**75GF:25CF**	**50GF:50CF**	**25GF:25CF**	**CF (%)**	**(VG)**	**FL (mm)**	**FD Glass Fiber (µm)**	**FD (µm) Carbon Fiber**	**MS (kN)**
Mean	5.2407	0.5556	0.4167	0.4074	0.4722	0.4259	10.0000	12.0000	11.9444	3.8889	13.7028
Standard Error	0.0745	0.1633	0.1359	0.1380	0.1474	0.1357	0.0000	0.0000	0.8298	0.2855	0.4240
Median	5.2500	0.0000	0.0000	0.0000	0.0000	0.0000	10.0000	12.0000	15.0000	5.0000	13.6700
Standard Deviation	0.5472	1.2001	0.9988	1.0144	1.0834	0.9972	0.0000	0.0000	6.0980	2.0982	3.1161
Standard Variance	0.2994	1.4403	0.9976	1.0290	1.1737	0.9944	0.0000	0.0000	37.1855	4.4025	9.7102
Range	1.5000	4.0000	4.0000	4.0000	4.0000	4.0000	0.0000	0.0000	15.0000	5.0000	14.6600
Minimum	4.5000	0.0000	0.0000	0.0000	0.0000	0.0000	10.0000	12.0000	0.0000	0.0000	5.1900
Maximum	6.0000	4.0000	4.0000	4.0000	4.0000	4.0000	10.0000	12.0000	15.0000	5.0000	19.8500
Confidence Level (95.0%)	0.1494	0.3276	0.2726	0.2769	0.2957	0.2722	0.0000	0.0000	1.6644	0.5727	0.8505

**Table 11 materials-15-08944-t011:** Performance evaluation of GP, M5P and RT model.

Models Approaches	CC	R^2^	MAE (kN)	RMSE (kN)	RAE (%)	RRSE (%)
**Training**						
**GP-PUK**	0.8383	0.7027	1.4276	1.7688	57.55	56.21
**M5P**	0.8396	0.7049	1.3358	1.7138	53.85	54.46
**RT**	0.8414	0.7079	1.2008	0.0171	48.41	54.42
**Testing**						
**GP-PUK**	0.8187	0.6702	1.5350	1.8524	64.56	59.57
**M5P**	0.8172	0.6678	1.5264	1.8331	64.20	58.94
**RT**	0.7936	0.6298	1.6573	1.9848	69.70	63.82

**Table 12 materials-15-08944-t012:** Performance evaluation ANN and MLR model.

Models Approaches	CC	R^2^	MAE (kN)	RMSE (kN)	RAE (%)	RRSE (%)
**Training**						
**ANN**	0.8858	0.7846	1.4449	1.8391	58.25	58.44
**MLR**	0.7647	0.5847	1.6509	2.0278	66.55	64.44
**Testing**						
**ANN**	0.8392	0.7042	1.4996	1.8315	63.07	58.89
**MLR**	0.7976	0.6361	1.6387	1.8910	68.92	60.81

**Table 13 materials-15-08944-t013:** Quartile values using actual and predicted values of all applicable models for the testing stage.

Statistic	Actual	ANN	GP-PUK	M5P	RT	MLR
Minimum	5.190	5.701	8.986	8.672	8.310	8.854
Maximum	19.850	16.941	16.802	16.724	18.530	16.741
1st Quartile	12.178	11.455	13.656	13.240	12.415	13.474
Mean	13.703	13.028	14.148	14.142	14.208	14.025
3rd Quartile	15.995	15.484	15.901	16.485	16.060	15.618
IQR	3.818	4.029	2.245	3.246	3.645	2.144

**Table 14 materials-15-08944-t014:** Feature importance analysis (MLR model).

Row No.	Input Parameter										Output Parameter	MLR Model		
												CC	MAE	RMSE
	BC (%)	GF (%)	75GF: 25CF	50GF: 50. CF	25GF: 75CF	CF (%)	Bitumen grade (VG)	FL (mm)	FD Glass (µm)	FD Carbon (µm)	MS (kN)			
1											*-*	0.8392	1.4996	1.8315
2												**0.7243**	**1.7134**	**2.1435**
3												0.7522	1.7549	2.0623
4												0.7578	1.7087	2.0387
5												0.7859	1.6874	1.9425
6												0.7859	1.6989	1.9439
7												0.7872	1.6408	1.9373
8												0.7963	1.6550	1.9059
9												0.7937	1.6628	1.9100
10												0.7941	1.6635	1.9200
11												0.7909	1.6739	1.9251

## Data Availability

Not applicable.

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
