# Peer review of "Marshall Stability Prediction with Glass and Carbon Fiber Modified Asphalt Mix Using Machine Learning Techniques"

_materials, 2022, doi:10.3390/ma15248944_

Round 1

Reviewer 1 Report

This is an interesting (for me) application of machine learning to model the properties of asphalt mixes, of great practical importance to industry. The paper is quite well written but has excessive detail in some places and needs to be checked for spelling, grammar, and typographical errors (e.g., line 346 RMSE = (1.7688, 18524)

The use of training and test sets is good but sparse feature selection methods like LASSO or MLR with expectation maximisation may generate sparser models with similar accuracy and easier interpretability

The bottom line is that all ML methods generate models of similar quality as shown by RMSE and MAE values in Table 11. Maybe just one result (the most interpretable one) could be used in the body of the manuscript and the other model results put in SI? The SVM seems to be overfitted comparing the training and test set results, this is often an issue with SVM (the Bayesian version, the relevance vector machine RVM, is preferred for that reason).

It would be helpful for add a multiple linear regression model results, preferably a sparse model like LASSO or MLREM. This would make feature importance easy to determine if the linear model is as good as the ML models, and would identify the effects of nonlinearity if it is worse.

The authors need to use an appropriate number of significant figures when quoting model outputs e.g., two decimal places for r2 values MAE and RMSE and zero for RAE and RMSS.

Figure 9, Table 9, Table 12 and related less relevant materials should be moved to SI as they distract from the story in the body of the paper and do not add much value there.

What is the large outlier with a measured value of 23 kN in the training set?

The model interpretation (feature importance) doesn't recognise that for nonlinear models, feature importance is a local rather than global property. Its value depends on where on the response surface of the model the feature importance is measured.

Reviewer 2 Report

Review comments

=============

This paper analyses different machine learning algorithms to determine the optimal algorithm for predicting the marshall stability of glass and carbon fiber-modified asphalt mix. The topic is quite interesting and overall, the manuscript is good and well-structured. However, the manuscript needs minor revision before acceptance for publication.

Specific comments

=============

Minor comments

---------------------

1.      The abstract should be able to tell what the research covers. The standard structure of the abstract should be maintained: “Introduction”, “Method”, “Results”, and “Conclusion”. The introductory part of the abstract should describe a bit of the problem the authors are trying to solve.

2.      The abstract should be revised for proper use of punctuation and missing punctuation.

3.      Before an abbreviation is used, the whole meaning should be stated first and then the acronym can be used.

4.      The authors should clearly state the novelty of their work or the work mainly about applying existing machine learning models to predict Marshall Stability?

5.      The authors should state the limitation of their work. What challenge was encountered during the study? 

6.  The authors should state the direction of their future work

Reviewer 3 Report

In this paper the authors propose a new method multiple machines learning algorithms, including artificial neural networks, support vector machines, Gaussian processes, and M5P tree-based models, were evaluated to determine the optimal algorithm for predicting the Marshall Stability of a glass and carbon fiber modified (hybrid) asphalt mix., while the original idea is interesting, the proposed measure has the room to be improved before the acceptance of the manuscript. Careful revision of the manuscript is necessary for its publication:

1.  Keywords must reflect the core of study same as abstract

2.Introduction should be clearly presented to highlight main ideas and motivation behind the

proposed research. Please include and clearly state research question and motivation of proposed

study in Introduction. The author should be covering the research gap.

3. the authors should analyze how to set the parameters of the proposed methods in the framework.  Do they have the “optimal” choice?

4.Figure and some tables captions need to be expanded to make them self-explained.

5.It would be good to have an overview of the proposed framework (preferably a system diagram or flowchart), to give readers a quick understanding of the framework.

6. Section experiment, it would be good to have more information about how experiments have been conducted. What tools/software has been used?

7.The discussion section in the present form is relatively weak and should be strengthened with more details and justifications.

8. Paper's writing, and grammar needs substantial work. English should be corrected by a native speaker prior to resubmitting.

9. The Literature citation is not adequate, and the related work to the machine learinng should be discussed:

9.1. Dual Regularized Unsupervised Feature Selection Based on Matrix Factorization and Minimum Redundancy with application in gene selection (2022),

9.2. Robust graph regularization nonnegative matrix factorization for link prediction in attributed networks (2022).

9.3. MVDF-RSC: Multi-view data fusion via robust spectral clustering for geo-tagged image tagging (2021)

Reviewer 4 Report

1. Title needs to be more clear. Marshall Stability Prediction using ANN, SVM, GP and M5P 2 Tree with Glass and Carbon Fiber Modified Asphalt Mix. 

You can refer to the ML approaches 

2. Abstract needs to be more concise. Only sensitivity analysis is included

3. As highlighted into the introduction, these are ML techniques. Further, you need to be clear in the introduction that why only the selected algorithms are used for this study

4. Literature is very generic like a thesis report and same like introduction. It needs be be critical literature of how others are using these machine learning model inline with the current title of research. Limit definitions of algorithms. Provide actual working from the prior work. 

5. Provide after the literature a comparison table of the approaches used by previous authors. You need to justify that why the algorithms you mentioned are only used in the study. 

6. "The materials used to conduct the experiments included bitumen (VG10) glass fibe rcarbon fiber, and filler, as well as open-graded coarse aggregates of 20 mm nominal size". Why these experiments and 20mm nominal size. There is no background information provided about these and directly mentioned in the methodology section

7. No clear methodology provided. How the study is conducted ? what are the variables ? the steps ? the data information ? All this information is extra information from 3.1 including section 4. Heading 4.1 provides the details of the dataset, that is the actual point from where authors should provide from where you are getting the data set. Now inside the collection of dataset, For the prediction of MS four ML techniques i.e., (ANN, SVM, GP and M5P Tree) were 288 implemented using Weka 3.9.5. This should be part of proper methods based on the prior literature. 

I am concerned about Weka as this tool is just for basic usage and there is no actual working of code on this software. Why Weka ?. The methodological aspects lack clarity

8. Now results and discussion have direct performance assessment without prior explanation and steps for the data collection. Needs to have proper logical flow of information

9. Figures have no explanation 

10. Only sensitivity analysis ? why. That also for ANN model. It is mentioned as best model. But not clear. As you have mentioned "actual and predicted outcome are within the ±25% error range in 323 both stages using ANN-based models.". Same in SVM: " depicts that the most of the predicted values fall within the margin error of ± 25%". Same in GP. most of the predicted values fall within the ±25% error range, which is shown by the agreement line that connects the actual and the predicted values". Then how ANN is better?. Also, all results look same. This means that data is not accurate or is biased

11. Comparison of applied models ?. is this research about ML or applied ?. The discussion is totally separate from the prior work. Results does not provide value for any algorithms

12. Very generic conclusion. Needs to be more details and future aspects.

Round 2

Reviewer 1 Report

The authors have addressed most of the concerns of this reviewer.

A number of important corrections and suggestions still need to be implemented before the paper is publishable. I would like to see the paper again to ensure these important changes have been made.

The NN and MLR models are very good, and the other ML models are equally so. The main issue is the paper is still much too long and complicated, with a lot of unnecessary information and analysis that confuses rather than clarifies the results of the modelling.

1. The following section in the Introduction is out of place and repeats much of the last paragraph in this section. Please delete "The current study focused on the prediction of Marshall stability of asphalt mixes constituted of glass, carbon, and glass carbon combination fibers by applying five machine learning models i.e., Artificial neural networks, Gaussian processes, M5P, Random tree and Multiple linear regression model 58 and further to determine the optimum model suitable for prediction of the Marshall stability in hybrid asphalt mixes. Equally important was to determine the suitability of each mix for flexible pavements. The criteria for selecting the said five machine learning techniques was their veracity in their functions".

2. Table 10 should be in Methods section in neural network subsection.

3. The metrics used to judge model quality could be limited to r2 (square of CC), RMSE and MAE.  These metrics should be limited to 2 decimal places as adding additional decimals is meaningless and confuses comparisons. The other metrics don't add anything to the understanding

4. Based on the test set RMSE and MAE in Table 11, NN, GP-PUK, and M5P have essentially the same predictive power and MLR and RT are only slightly worse. I would only report the NN and MLR results in the body of the paper and consign the other results to Supplementary Information (including Figs 6-9). This will simplify the paper and not confuse the discussion by talking about other Ml results that are essentially the same. The reason for keeping the MLR in the body of the paper is that this model is very easy to interpret. The second Fig. 9 and Fig.s 10,11 should be removed or moved to Supplementary Information as they don't display anything useful.

5. The sensitivity analysis section should be renamed Feature Importance and the MLR equation only be analysed for feature importance. the slight nonlinearity of the problem identified by the NN model being slightly better than the MLR model makes feature importance complex. Analysis of the MLR equation would be more than adequate for the paper and would identify the sign and magnitude of the contribution of each input feature to the asphalt properties.

6. You have two Fig 9s. 

7. The full dataset must be provided in the Supplementary Information or on an accessible data archive site.

Reviewer 3 Report

I have gone through the revised paper. All my concerns and requests have been carefully addressed by authors.

Reviewer 4 Report

Dear Authors,

thank you for the submission. I can see that you have now done a good job and worked on various sections. Good efforts for improvement. The manuscript is now ready for the publiation.

Thank you

Round 3

Reviewer 1 Report

The authors have implemented most of the suggestions from the last review round.  They are still not using too many significant figures for r2, RMSE an MAE in some places, which should be corrected.

The most important issue that must be corrected is in section 8 relating to feature importance.  The authors should provide the MLR equation in the paper e.g.,BC (%)

MS = a x BC% + b x GF% + c x 75GF: 25CF + d x 50GF:50CF + e x 25GF:75CF + f x CF(%) + g x Bitumen grade + h x FL (mm) + i x FDGlass + j x FD Carbon + k

The coefficient of each input parameter (descriptor or feature, a-k above) tells you the sign and the magnitude of the contribution of each feature to the modelled asphalt property. They could also provide a table of MLR coefficients for each feature and base their description of feature importance around these. The current feature importance section is very uninformative. 
